# Use of Multifactorial Treatments to Address the Challenge of Translating Experimental Myocardial Infarct Reduction Strategies

**DOI:** 10.3390/ijms20061449

**Published:** 2019-03-22

**Authors:** Julie L. Horton, Jitka Virag

**Affiliations:** 1Burdock Group, 859 Outer Road, Orlando, FL 32814, USA; juliehortonphd@gmail.com; 2Department of Physiology, Brody School of Medicine, 600 Moye Blvd, East Carolina University, Greenville, NC 27834, USA

**Keywords:** myocardial infarction, cardioprotection, therapeutic strategies, ephrinA1, ischemia, reperfusion

## Abstract

Myocardial tissue damage that occurs during an ischemic event leads to a spiraling deterioration of cardiac muscle structural and functional integrity. Reperfusion is the only known efficacious strategy and is the most commonly used treatment to reduce injury and prevent remodeling. However, timing is critical, and the procedure is not always feasible for a variety of reasons. The complex molecular basis for cardioprotection has been studied for decades but formulation of a viable therapeutic that can significantly attenuate myocardial injury remains elusive. In this review, we address barriers to the development of a fruitful approach that will substantially improve the prognosis of those suffering from this widespread and largely unmitigated disease. Furthermore, we proffer that ephrinA1, a candidate molecule that satisfies many of the important criteria discussed, possesses robust potential to overcome these hurdles and thus offers protection that surpasses the limitations currently observed.

## 1. Introduction

Cardiovascular disease (CVD), a general term for heart dysfunction resulting from a wide range of etiologies, is a significant health concern that has been the leading cause of death contemporaneously and in recent history [1]. A heart attack, or myocardial infarction (MI), is one of the most common occurrences that can either result from or cause CVD. According to the Fourth Universal Definition of Myocardial Infarction, MI is defined as myocardial tissue injury detected by divergent cardiac biomarkers and accompanied by evidence of ischemia. The cause of the injury does not preclude designation as an MI [2]. In a clinical setting, patients with an MI are assessed to determine the degree of the occlusion. A complete blockage is designated as an ST-segment elevation myocardial infarction (STEMI) and represents the most serious type of MI. A heterogeneous level of blockage is termed a non-ST-segment elevation myocardial infarction (NSTEMI) [3].

MI is a significant international health problem. Globally, the World Health Organization estimates over 32 million MIs and strokes occur annually [4]. Notably, precise quantification of the mortality attributable to MI is complicated by the lack of a universal standardized method to designate and record MI related deaths. Despite this difficulty, the Global Health Data Exchange (GHDx) conducts an annual Global Burden of Disease (GBD) study using the available international data. The most recent GBD study grouped MI related deaths under an umbrella cause of death term- “ischemic heart disease” (IHD). The analyses of the data collected for this GBD study showed that IHD was a contributing factor in almost 16% of deaths worldwide. These analyses also examined regional, socioeconomic, and other demographic categorizations, and a subset of this data is shown in Table 1 [5]. 

An annual report published by the American Heart Association (AHA) gives a more granular view of the incidence and mortality associated with MI in the United States (US). In the US over 750,000 adults per year are afflicted with an MI [6]. Furthermore, the survival rate of US adults that have an out-of-hospital cardiac arrest (OHCA) and subsequently receive emergency medical services for the MI is an abysmal 10–11% [7]. The high mortality rate in persons that experience an OHCA epitomizes the urgent need for development of novel, acute therapeutic strategies to prevent or repair myocardial tissue damage resulting from an MI. Moreover, the health and well-being, productivity, longevity, and economic impacts of this irreversible condition are projected to worsen. There is also significant variability in the incidence, prevalence, and mortality associated with MI with respect to gender, ethnicity, comorbidities, socioeconomic status, and age [7]. Thus, more effective, personalized strategies for reducing infarct size and ultimately reducing mortality from MI should be vigorously pursued.

The scientific community is cognizant of the need to reduce MI-mediated tissue injury and its impact on global cardiac function as evidenced by the considerable amount of research over the last 40 years dedicated to understanding the mechanisms by which cardiac tissue can be preserved in the face of such stress (Figure 1). A wide range of experimental paradigms in various in vivo and in vitro animal and cell culture models have led to the discovery of over 100 mediators of preconditioning and postconditioning that can be affected both locally and remotely. Many pathways and molecules ranging from lipids, hormones, ion channels, cytokines, enzymes, transcription factors, mRNAs, and subcellular organelle function/production have been demonstrated to be involved in one or more processes that provide cellular protection from ischemia. To-date however, there have been no formulation/administration of these mediators that have successfully reached the clinical realm.

The morbidity and mortality suffered as a consequence of MI are positively correlated with infarct size. The region of heart affected is also related to the degree of morbidity and rate of survival from an MI because valve function may be compromised [8,9]. A clear threshold designating an infarct size that definitively improves survivability has not been established. However, a systematic analysis of data pertaining to infarct size, ejection fraction, and the rate of mortality following an MI demonstrated that achieving an infarct that is <20% of the left ventricle (LV) would likely result in an improved outcome [10]. Indeed, there are current pharmacotherapies that are unconventional treatments for MI but show evidence of improving the outcome of MI due to decreased scar formation and/or a reduction in infarct size.

Ivabradine is currently used in a clinical setting as an antianginal agent and is effective at reducing heart rate [11,12,13]. Ivabradine has also been shown in rodent models of MI to significantly reduce scarring [14,15]. Another approach to reducing scar tissue in the infarcted heart is coaxing stem-cells to the damaged area. To this end, granulocyte colony stimulating factor (G-CSF) has been shown to attract stem cells from the bone marrow to damaged cardiac tissue and subsequently induce differentiation into mature cardiomyocytes and endothelial cells [16,17,18]. Metoprolol, a conventional β-blocker, has been shown to result in decreased infarct size when administered within a specific timeframe after the onset of ischemia [19,20].

The current treatment for MI depends on whether the patient is given a diagnosis of STEMI or NSTEMI [21]. A patient presenting with STEMI is always first prescribed reperfusion therapy. An initial diagnosis of NSTEMI, though, may result in a variety of alternative non-invasive or minimally invasive treatments to help ease the blockage [22]. This review focuses on addressing the unmet needs for therapies to improve STEMI outcomes given that a STEMI results in rapid, irreversible damage to cardiac tissue. It is widely accepted that reperfusion therapy given to a STEMI patient can generally slow the progression of heart failure. In the long-term, though, reperfusion does not restore healthy cardiac functionality or make significant gains in long-term survival [10]. This lack of efficacy is, at least in part, due to the time-sensitive nature of reperfusion therapy and the fact that on average reperfusion does not reduce infarct size to < 20% of the LV. In fact, reperfusion therapy results in a median infarct size of 50% [10,23,24]. 

There are many obstacles currently blocking the progress of treating MI in humans. For example, pre-clinical models of MI largely fail to sufficiently replicate the conditions of human disease because many pathologies are derived from complex etiologies and commonly occur as comorbidities in the human patient population [25,26,27]. Concerns regarding time, dose/frequency, and delivery mode also remain significant impediments to translation of candidates that have demonstrated pre-clinical potential [28,29,30]. Notably, though, some recent advancements with regards to delivery were reported. After a 32 month duration, follow-up with patients given percutaneous coronary interventions (PCI) incorporating the use of second-generation drug-eluting stents (DES) were found to have comparable outcomes to patients that received coronary artery bypass surgery (CABS) [31]. It remains to be seen if PCI with second-generation DES is equally effective in the long-term and in patients with specific pathologies. Long-term studies (5 years) comparing PCI incorporating first-generation DES to CABS found a significant increase in major adverse cardiac and cerebrovascular events in the PCI cohort compared to the CABS cohort [32]. In a long-term study (5-years) diabetic patients with a DES implantation did not show any difference in rates of death, MI, or stroke than diabetic patients that received CABS. However, the DES group did have a significantly higher rate of repeat revascularization [33].

Recent research also suggests emerging therapies in the form of nutritional intervention for improving the outcome of heart failure. The heart undergoes a metabolic phenotype shift during the progression from ischemic insult to decompensated hypertrophy. The normal, healthy heart overwhelmingly uses fatty acid oxidation (FAO) to generate ATP, which is the energy currency for contraction [34]. However, when a pressure-overload stress occurs, as is the case with strain exerted on surrounding cardiac myocytes from damaged myocardial tissue, the reliance on FAO to generate ATP is decreased [35,36]. Recently, it was reported that under stressed conditions, like pressure overload, the heart increasingly relies on ketone bodies as a fuel substrate [37]. Furthermore, the increased rate of ketone oxidation was found to be an adaptive mechanism and when ketones were increasingly made available to the heart, the function of the failing heart was significantly improved [38].

Overall, analyses of cardioprotective strategies employed thus far has propelled a hypothesis that a singular targeted approach to treating myocardial ischemia-reperfusion injury is insufficient, and a multifactorial treatment paradigm will provide better outcomes for patients [25]. One way of implementing multifactorial treatment is to use a combination of therapies that have had at least some success in the clinical realm. Indeed, this has been a successful approach in many trials thus far and is reviewed in great detail elsewhere [25]. Another way to address the need to treat MI with a multifactorial approach, is to use a single molecule that targets multiple pathways. One such intervention that shows promise by conferring a benefit to the outcome of ischemia/reperfusion is treatment with ephrin A1-Fc (EA1-Fc). 

## 2. Pathophysiology of MI

When coronary occlusion occurs, cardiac myocytes are starved of oxygen. These terminally differentiated cardiomyocytes become ischemic and quickly begin to undergo necrosis, apoptosis [39,40,41], and/or autophagy [41,42,43,44]. The damage from an MI is rapid, with necrosis occurring within the first 40 min of blockage [42]. Despite the controversy surrounding the potential for cardiac repair using endogenous or exogenous progenitor cells, significant obstacles regarding host integration, clinical feasibility, and functional outcomes must be overcome. As such, once the necrotic tissue is removed, fibroblasts deposit a non-contractile matrix of collagen, leading to scar formation. The scar tissue is rigid and, combined with increased fibrosis in the viable portions of the myocardium, impairs both the contractility and relaxation of the heart muscle, i.e., impaired diastolic filling and systolic ejection [41,45,46,47,48]. 

To make matters worse, the systemic physiological response to ischemic stress causes further damage. The immune system responds immediately to ischemia in the heart with neutrophils infiltrating the oxygen-deprived tissue. Shortly after neutrophils have migrated to the ischemic zone, leukocytes (mostly macrophages) arrive and begin to facilitate digestion of necrotic cells [39,41,46,49,50,51]. While the immune system is critical for tissue repair in terms of remodeling to promote revascularization and scar formation, the neutrophils also release reactive oxygen species (ROS), cytokines and chemokines, and proteolytic enzymes, which cause further damage to surrounding cells and the intercellular matrix. Ultimately, this systemic response to ischemia backfires and causes a larger infarct zone [41,42,45,46,47,48].

The fibrotic scar that replaces lost contractile myocardium following an ischemic event undergoes continuous remodeling. The extracellular matrix (ECM) degrades in response to increased matrix metalloproteinase (MMP) activity and the fragments generated play a role in propagation of the inflammatory response as well as the deposition of replacement scar tissue, which itself evolves from the early fibrin-based matrix to a collagen-based mature scar [52]. The dynamic changes in composition of the ECM that occur in response to the polarization of immune cell subsets from the pro-inflammatory to the resolution phase is also complex, as is the sequelae of fibroblast activation and scar formation, both of which remain under intensive investigation [53,54,55,56,57,58]. Of note, the juncture between the non-contractile scar and contractile viable myocardium creates strain, which, in animal models with large MIs, is predisposed to rupture. The beat-to-beat tension placed on this region is not often considered but requires special attention [59]. The scarred heart is subject to increased workload to sustain adequate cardiac output. Under increased stress, the healthy, non-proliferative cardiac myocytes also begin to hypertrophy leading to LV remodeling and subsequent chamber dilation. The hypertrophied cardiac myocytes further exacerbate the already strained heart, and this, combined with increased interstitial fibrosis, results in eventual inability for the contractile apparatus of the heart to contract strongly enough to maintain sufficient circulatory blood flow (i.e., heart failure) [41,60,61,62,63]. Without sufficient blood flow, systemic oxygen deprivation leads to organ failure and eventual death. Thus, if the initial infarct injury can be reduced, the downward spiral to heart failure and mortality may be at least partially averted. Additionally, reductions in the initial infarct injury result in increased preservation of 3D tissue structure, which is beneficial for surgical reconstruction [64].

## 3. Ephrin A1 Ligand Intramyocardial Injection is Cardioprotective

One of the most commonly employed strategies to develop a new therapeutic target is to investigate what changes occur on a molecular level between the healthy and diseased state. To this end, Dries, Kent, and Virag (2011) found that the gene expression profile of the infarcted heart changed dramatically when compared to the healthy heart. Some of the changes that occurred in this study included altered expression of ephrinA1 (EA1) and several EphA receptors. Transcript levels of EA1 were reduced by 35% after MI while expression of Eph receptors A1, A3, and A7 were increased [41]. The observed opposite directionality of expression changes strongly suggested decreased activation of the Eph receptors in the infarcted myocardium. 

The hypothesis that decreased cardiomyocyte EA1 expression was a contributing factor to the pathology of MI was tested by direct intramyocardial injection of recombinant EA1-Fc ligand into the heart immediately following permanent occlusion of the left anterior descending coronary artery. When EA1-Fc was administered at the time of MI onset, the damage to the heart was significantly reduced at 4 days post-MI, leading to reduced infarct size, less necrosis, less ventricular dilation, and less inflammation [41]. Histological examination revealed that the levels of endogenous EA1 in the infarcted heart were situated exclusively in the healthy regions of the heart tissue including the border zone of the infarcted region. Interestingly, in hearts treated with EA1-Fc, 4 days post-MI, endogenous EA1 protein expression appeared in the infiltrating granulation tissue cells throughout the region of the infarct. The levels of endogenous EA1 were maintained at levels present in healthy myocardium when EA1-Fc is provided immediately following MI as well [41].

This effect combined with decreased endogenous EA1 in infarcted myocardium, suggests that EA1-Fc is a cardioprotective molecule [41]. In addition to direct positive effects on the infarcted region and surrounding myocardium, EA1-Fc administration directly after MI also improves biomarkers of MI severity. One such biomarker, cardiac troponin I (cTnI) is a well-known correlate to the size of MI injury [41,65,66,67,68,69,70]. In fact, the clinical diagnosis of MI includes serum levels of cardiac troponin exceeding the 99th percentile upper reference limit (URL) [2,71]. When cTnI values exceed the 99th URL and fluctuate (rising and/or falling), the injury to the myocardium is considered acute. Conversely, if there is <20% fluctuation in cTnI values, the injury to the myocardium is considered chronic [2]). When EA1-Fc is injected acutely into the infarcted heart, levels of cardiac troponin I are decreased by approximately 55%. In this study, cTnI levels were only assessed at one point in time (4-days post- surgery and treatment) [41]. Studies to measure cTnI levels at various points in time are needed to further elucidate the mechanism of benefit conferred by EA1 treatment.

Additional experiments conducted in an acute I/R model further support the hypothesis that EA1 is a cardioprotective agent. Specifically, measurements of tissue viability and infarct regions revealed a marked drop in necrotic tissue in EA1-Fc treated mice in response to acute ischemia and reperfusion (30min I/24hr R) compared to vehicle control mice [42]. Analysis of heart function using M-mode echocardiography showed that I/R mice treated with IgG-Fc control, had significantly lower fractional shortening (FS) and ejection fraction (EF) compared to the sham controls. In contrast, mice injected with EA1-Fc did not have significantly different FS or EF measurements compared to the uninjured sham controls. Four days post-MI, vehicle control mice displayed further decreases in FS and EF measurements indicating worsening heart function over time. Again, even after 4-days of reperfusion, the I/R mice provided EA1-Fc injections did not have significantly different FS and EF compared to the sham controls [42]. In addition to EA1, other ephrin ligands and receptors (B family) have been reported to play a cardioprotective role as well [72], but there have been no further investigations. The unique bidirectional signaling system combined with the potential expression of multiple receptor subtypes in each cell type in cardiac tissue adds considerable complexity to deciphering the mechanisms by which protection occurs. The potential for this system to be exploited for the simultaneous attenuation/stimulation of multiple pathways that culminate in reduced injury and deleterious remodeling remains under investigation. 

## 4. Potential mechanisms for Ephrin A1-mediated cardioprotection

Substantial information about the physiological ephrin ligand-receptor interaction has been uncovered in the last 30 years. Since ephrin tyrosine kinase receptor was discovered in 1987 [73] research regarding the ephrin ligand-receptor interaction indicates characteristic bidirectional signaling [41,74,75,76,77]. Once ephrin ligands bind their respective receptors, the receptor undergoes autophosphorylation. Phosphorylation of the ephrin receptor then triggers endocytosis and subsequent degradation of the ligand–receptor complex [78]. The administration of chimeric EA1-Fc in MI has a profound effect on expression of ephrin ligands and receptors, which may provide insight into the mechanism of cardioprotection. The potential effect of EA1 signaling on the healthy and infarcted myocardium is summarized in Figure 2. 

Eight EphA receptors are expressed in mice and two of these, EphA5 and EphA8, are not expressed in the healthy mouse myocardium. Of the six Eph receptors expressed in healthy myocardium, EA1-Fc affects expression levels of five of these (A1, A2, A3, A4, and A6), and as mentioned previously, positively regulates endogenous EA1 ligand expression [41,42]. Ephrin ligands A2-A5 and ephrin B3 are expressed in the normal heart but levels of expression do not change in response to MI or EA1-Fc treatment [41]. Ephrin signaling is known to be involved in a variety of cellular processes including growth, differentiation, motility, and survival [41,77,78,79,80,81]. Notably, EA1 is the only ephrin ligand that binds to all eight ephrin receptors expressed in mice, so all of the cellular processes influenced by ephrin receptors, which are expressed on each cell type, are potential candidates for the mechanism behind EA1-Fc-mediated cardioprotection [41].

For example, there is strong evidence to suggest that EA1-Fc administration reduces apoptosis in the infarcted heart. The amount of cleaved PARP with EA1-Fc treatment in MI is decreased compared to the MI hearts treated with vehicle control [41]. Levels of cleaved PARP are positively correlated with apoptosis [82,83,84]. Also, in this study Bcl-2-associated athanogene-1 (BAG-1) protein, which is known to enhance Bcl-2 anti-apoptotic effects [85,86], had expression levels increased by approximately 54% in the EA1-Fc treated group. EA1-Fc administration also resulted in increased phosphorylation of AKT [41]. AKT signaling is known to regulate myocyte survival with an increase in pAKT/AKT being indicative of increased survival [87,88,89,90].

Another possibility for consideration is that the benefit conferred by EA1-Fc administration may be attributed to the role of ephrin signaling in autophagy. EA1-Fc treatment resulted in an increase of LC3II/LC3I ratios and decreased pmTOR/mTOR ratios compared to IgG-Fc treated MI counterparts [42]. The change in both of these ratios suggest an increase in autophagy, which would carry the benefit of repairing damaged tissue as this process removes dysfunctional cellular components and recycles them to produce energy [91,92,93,94,95].

Additionally, EA1 signaling has been shown to have a role in angiogenesis primarily by mediating endothelial cell migration [41,96,97,98]. EA2, EA6, and EA7 receptors are also expressed on endothelial cells and have been implicated in angiogenesis [41,99,100,101,102]. However, an angiogenic response was not apparent in the EA1-Fc injected MI heart. No differences were observed in endothelial cell proliferation or capillary density between vehicle controls and EA1-Fc treated hearts [41]. 

It is known that ephrin receptors are expressed at different levels in early and late stages of inflammation, suggesting a role for these receptors in the inflammatory response [41,103]. Indeed, it does appear that EA1-Fc does affect the inflammatory response in MI. Compared to vehicle treated controls, EA1-Fc injected hearts had a 57% reduction in neutrophil density and a 21% reduction in leukocyte density. Also, EA1-Fc injection reduced protein levels of nuclear factor *κ* light-chain enhancer of activated B cells (NF-*κ*B) [41]. NF-*κ*B is known to be involved with mediating inflammation [49]. Taken together, these studies demonstrate robust multitarget properties of ephrinA1-Fc that coordinately reduce injury and promote tissue survival.

## 5. Conclusions

Despite the massive need and corresponding effort exerted, finding a practical and efficacious agent for cardioprotection has proven to be an elusive task. In 2010, the National Institutes of Health established a “consortium for preclinical assessment of cardioprotective therapies”, otherwise known as CAESAR to address the longstanding futility surrounding cardioprotection. The rationale was to establish an infrastructure that would increase the rigor of scientific research such that proposed cardioprotective molecules would likely succeed in efficacy [104]. However, despite this conceptually valuable paradigm shift, no overtly effective, clinically relevant, cardioprotective agents have emerged. 

Numerous factors likely contribute to the lack of progress with regards to rehabilitating damaged myocardium. One factor may be that the treatment needs to be multi-faceted [25,105]. The majority of known cardioprotective factors only influence one or two aspects of tissue injury and recovery [30,106,107]. One study examined the effect of applying a combination of therapies to the outcome on infarct size. This study found that using a combination of three major MI treatments showed a 55% decrease in infarct size, far exceeding benefits typically obtained by a single therapeutic intervention [108]. This combination treatment strategy is similar to the benefit observed with a single EA1-Fc treatment. Acute treatment of MI with EA1-Fc ligand has similar profound effects in preclinical studies. An average reduction of 50% infarct size comes from a single EA1-Fc treatment alone at the time of coronary artery ligation and complete functional recovery in reperfused myocardium [41,42] (Figure 3). 

The robust cardioprotection afforded by EA1-Fc treatment is in the early stages of research with respect to mechanism and duration of protection. Much work in the discovery and development phase of drug development remains before pre-clinical studies can begin. Of particular note, there are outstanding questions regarding the route of administration, timing, and dose of the EA1-Fc treatment. These issues must be investigated further before EA1-Fc can advance to pre-clinical pharmacological research. The method/substance applied must coordinately influence the cascades that promote protection and/or reduce injury at the cellular and molecular level of most, if not all, cell types involved. Given the controversy and multidimensional nature of questions surrounding the parameters of an optimal intervention, many unknowns still remain [25,29]. Additionally, there is still debate throughout the cardiac research community regarding what magnitude of difference is needed to establish efficacy and what endpoints should be used for this measure [109]. The traditional measurement of infarct size may not be the best indicator for extrapolating therapeutic potential due to differences in body size and heart rate. Instead, functional outcomes normalized to an uninjured control may be a better metric. Another hurdle that EA1-Fc must overcome is discovery of the mechanism by which it confers cardioprotection. It may be that EA1-Fc simply amplifies, inhibits, or maintains the normal activity of EphA receptors. The possibility also exists, though, that the addition of exogenous EA1-Fc ligand during the stress of I/R conveys its positive effects through an alternative mechanism. Ultimately, uncovering the mechanism of EA1-Fc-mediated cardioprotection would allow great insight into what processes can be targeted for future development and individualized MI treatments to optimize clinical outcomes.

## 6. Patents

The work references Patents 8,580,739 issued 11/12/13 and 9,974,831 issued 5/22/18. 

## Figures and Tables

**Figure 1 ijms-20-01449-f001:**
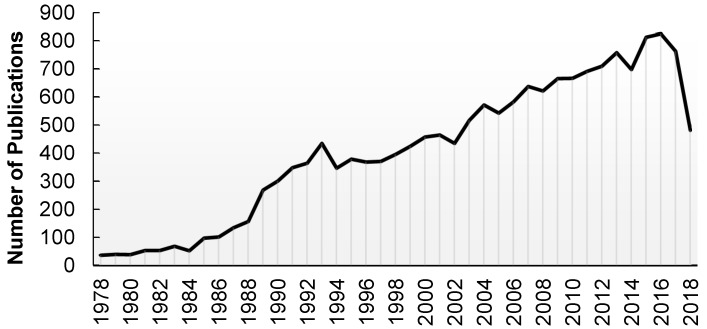
Literature search on Pubmed for the terms related to cardiac ischemia tissue injury show 16,752 publications since 1978. Search details: (“heart”[MeSH Terms] OR “heart”[All Fields] OR “cardiac”[All Fields]) OR (“ischaemia”[All Fields] OR “ischemia”[MeSH Terms] OR “ischemia”[All Fields]) OR (“tissues”[MeSH Terms] OR “tissues”[All Fields] OR “tissue”[All Fields]) OR (“wounds and injuries”[MeSH Terms] OR (“wounds”[All Fields] AND “injuries”[All Fields]) OR “wounds and injuries”[All Fields] OR “injury”[All Fields]) AND (“1978/01/01”[PDAT]: “2018/11/29”[PDAT]): https://www.ncbi.nlm.nih.gov/pubmed/; site visited on 29 November 2018.

**Figure 2 ijms-20-01449-f002:**
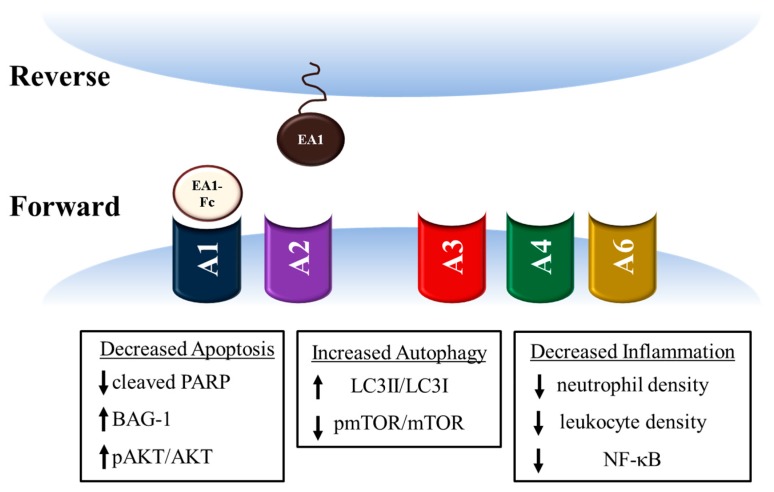
Potential Mechanisms of EA1-Fc Mediated Cardioprotection. Treatment with Ephrin A1-Fc (EA1-Fc) in a mouse model of myocardial infarction (MI) was shown to alter expression of five Ephrin receptors (EphA1-A4 and EphA6) as well as endogenous EA1 ligand when compared to the vehicle treated MI group. Canonical ephrin signaling regulates a wide-variety of physiological processes. The cardioprotective role of EA1-Fc may be attributed to ephrin signaling. In support of this, in the EA1-Fc MI group, indicators of decreased apoptosis, increased autophagy, and decreased inflammation were observed. The possibility also exists that EA1-Fc injection is cardioprotective through a mechanism that is not normally attributed to ephrin signaling.

**Figure 3 ijms-20-01449-f003:**
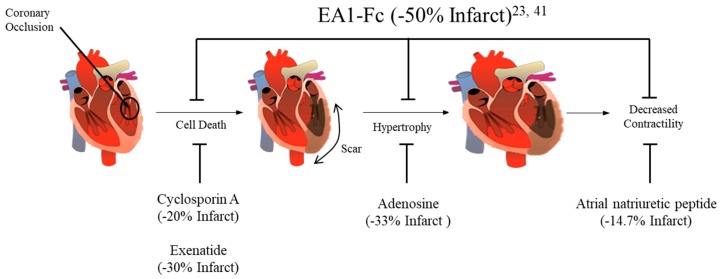
Comparison of pharmacological cardioprotective strategies’ efficacy to reduce infarct size due to ischemia/reperfusion injury [106].

**Table 1 ijms-20-01449-t001:** Ischemic heart disease related deaths in select demographics [5] *.

Grouping	% of All Deaths
Global	15.96266586
Low SDI	8.554976163
Middle SDI	16.63194299
High SDI	16.59093766
Mexico	14.2476225
United States	18.65732133
Southeast Asia	13.05553801
South Asia	15.47558271
Central Asia	33.26288153
Indonesia	14.29116844
Oceania	13.89558582
Australasia	16.54619353
Latin America and Caribbean	14.05392127
Tropical Latin America	13.07831696
Southern Latin America	14.71951694
Eastern Europe	34.96210128
Central Europe	26.35190592
Western Europe	15.93160107
United Kingdom	14.30616833
Sweden	19.92329938
Central Sub-Saharan Africa	5.589027664
Southern Sub-Saharan Africa	6.811094508
Sub-Saharan Africa	4.942820494
North Africa and Middle East	24.78925909
Western Sub-Saharan Africa	4.443263763
Eastern Sub-Saharan Africa	4.873600872

* Includes males and females of all ages. Shown as percent (%) of total deaths in group. SDI = socio demographic index.

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
