# Peer review of "Use of Multifactorial Treatments to Address the Challenge of Translating Experimental Myocardial Infarct Reduction Strategies"

_ijms, 2019, doi:10.3390/ijms20061449_

Reviewer 1 Report

General comments:

In this work, authors provide a well-rounded review article examining the contemporary problem of post-ischemic MI ventricular remodeling and discuss pathophysiology and mediators that contribute to infarct size development and challenges that need to be addressed in order to reduce infarct size and its unfavorable sequelae such as MI-related heart failure and systolic dysfunction. Authors emphasize their work on Ephrin A1 ligand and its receptors as a potential pharmacotherapeutic avenue in combating infarct size and possibly improving clinical outcomes.

 Specific comments:

-          Generally, Introduction is well-written, especially pathophysiology behind MI, with certain areas that could use some refinement, please see below:

-          Lines 29-32, etc. Please acknowledge in the text that MI is a complex entity that is been defined currently by the 4th definition of MI and likewise it can be caused by mechanisms beyond arterial blockage/atherosclerotic etiology

-          Introduction, you mention statistics in the US regarding MI per annum, what about European and/or worldwide statistics? Please elaborate.

-          Please substantiate infarct size and morbidity/mortality statements with appropriate references

-          In the Introduction section, could the authors elaborate briefly on the heterogeneity of NSTEMI vs. STEMI and emphasize reperfusion as a gold standard for STEMI whereas in NSTEMI there is a wide array of practices and if you could connect this with infarct size I think would be beneficial to this work

-          Line165-166. Please define troponin criteria for a diagnosis of MI precisely, as laid out in the 4th universal definition of MI

-          It would be beneficial to know in what time-frame are cTnI levels decreased upon infusing EA1-Fc. Is it in the second measurement? Is the peak troponin reduced? These need to be discussed. Are there any studies performed in humans or pre-clinical that use imaging upon infusion of EA1? Studies that assessed the viability of myocardium in terms of imaging studies, not only biomarkers of myocardial injury such as troponin? If these are available they should be included in this review and further discussed.

-          I would advise authors to provide a relevant schematic Figure summarizing the net effects of EA1 signaling and EA1-Fc on healthy and infarcted myocardium. Such a figure would improve this manuscript greatly and would enable better appreciation and understanding of physiological/pathophysiological implications of this molecule

-          Authors could provide a small/medium-size paragraph to current pharmacotherapeutic avenues that could limit scar formation/infarct size in the clinical setting as this would enhance the „clinical“ side of cardioprotection or non-dogmatic treatments that might influence these endpoints

-          Please consider references on, for example, ivabradine use in the ACS (Niccoli et al. PMID: 28256323; Priti et al. PMID 28423233; O'Connor et al. PMID: 27107133), granulocyte-colony stimulating factors (D'Amario et al. PMID: 28602846; Huang et al. PMID: 29651017), metoprolol (Garcia-Prieto et al. PMID 28416795; Pizarro et al. PMID: 24694530), etc.

-          Please check and fix references throughout the manuscript and at the References list as it seems some have missing information

Author Response

Dear Reviewers:

              We appreciate the comments you provided and believe the changes made based on your comments will significantly strengthen the manuscript entitled, “Use of Multifactorial Treatments to Address the Challenge of Translating Experimental Myocardial Infarct Reduction Strategies.” We have addressed your comments point-by-point in Italics in the following letter and included a “Tracked Changes” revised version of the document.

Reviewer 1:

General comments:

In this work, authors provide a well-rounded review article examining the contemporary problem of post-ischemic MI ventricular remodeling and discuss pathophysiology and mediators that contribute to infarct size development and challenges that need to be addressed in order to reduce infarct size and its unfavorable sequelae such as MI-related heart failure and systolic dysfunction. Authors emphasize their work on Ephrin A1 ligand and its receptors as a potential pharmacotherapeutic avenue in combating infarct size and possibly improving clinical outcomes.

Specific comments:

-          Generally, Introduction is well-written, especially pathophysiology behind MI, with certain areas that could use some refinement, please see below:

-          Lines 29-32, etc. Please acknowledge in the text that MI is a complex entity that is been defined currently by the 4th definition of MI and likewise it can be caused by mechanisms beyond arterial blockage/atherosclerotic etiology.

Thank you for the suggestion to include this more recent definition of MI. We updated the definition and source per your suggestion. The revised language is incorporated, and the reference has been updated accordingly. In addition to acknowledging the clinical complexity of MI in humans, we also added commentary on the different experimental models used in research to study MI.

-          Introduction, you mention statistics in the US regarding MI per annum, what about European and/or worldwide statistics? Please elaborate.

We looked through the literature extensively and came to the conclusion that there is quite a bit of discrepancy between regional definitions of MI and data collection to record incidence. To this end, we did not feel it was appropriate for us to make a direct quantitative comparison between the US incidence of MI annually and other regions. We did; however, try to highlight some of the key systematic analytical data on ischemic heart disease-related mortality. The new information is in paragraph 2 of the introduction and Table 1. We found it quite interesting that the less developed regions of the world seem to have lower % mortality than more developed countries. We did not articulate this observation in the review article because we feel there are numerous potential explanations for this discrepancy. For example, one potential explanation is that in less developed regions the mortality from MI is dwarfed by other causes of death (i.e. untreated infectious diseases) which are routinely treated in more developed regions.

-          Please substantiate infarct size and morbidity/mortality statements with appropriate references

Appropriate primary references were added and language was clarified(see paragraph 5 of introduction).

-          In the Introduction section, could the authors elaborate briefly on the heterogeneity of NSTEMI vs. STEMI and emphasize reperfusion as a gold standard for STEMI whereas in NSTEMI there is a wide array of practices and if you could connect this with infarct size I think would be beneficial to this work.

Thank you for the suggestion. We added information throughout the introduction pertaining to the differences in treatment modalities between a STEMI vs. NSTEMI as well as appropriate references.

-          Line165-166. Please define troponin criteria for a diagnosis of MI precisely, as laid out in the 4th universal definition of MI

Section 3 of the manuscript; paragraph 3 was revised to be more specific.

-          It would be beneficial to know in what time-frame are cTnI levels decreased upon infusing EA1-Fc. Is it in the second measurement? Is the peak troponin reduced? These need to be discussed. Are there any studies performed in humans or pre-clinical that use imaging upon infusion of EA1? Studies that assessed the viability of myocardium in terms of imaging studies, not only biomarkers of myocardial injury such as troponin? If these are available they should be included in this review and further discussed.

Regarding troponin- unfortunately, the EA1-Fc studies published to-date reflect the very early stages in a long process of investigation. For the cTnI levels, only one time point was measured (4 days post-surgery/treatment) in the cited experiment. We completely agree that there is much to be done to elucidate the mechanism by which EA1-Fc confers benefit and to further define circumstances that EA1-Fc treatment confers benefit. To this end, we added clarification in Section 3 Paragraph 3 to acknowledge the preliminary nature of the data.

As for endpoints other than cTnI measurements to address viability and function of the myocardium, we discuss the histological results reported (Section 3 paragraph 2) in the work cited that show EA1-Fc treatment results in reduced infarct size, less necrosis, less ventricular dilation, and less inflammation. Echo data is also discussed to address changes in functionality (Section 3 paragraph 4). Again, though, we do recognize (and hopefully make clear in the manuscript) that much work remains.

We are not aware of any published pre-clinical or clinical research that have utilized EA1-Fc. Our literature search only revealed studies in the discovery of small molecule antagonists for receptors as cancer therapeutic and early stages of the drug development process. We revised the conclusion paragraph 3 to clarify this point.

-          I would advise authors to provide a relevant schematic Figure summarizing the net effects of EA1 signaling and EA1-Fc on healthy and infarcted myocardium. Such a figure would improve this manuscript greatly and would enable better appreciation and understanding of physiological/pathophysiological implications of this molecule.

Thank you for the suggestion. We have added a figure to summarize the potential net effects of EA1-Fc signaling in the infarcted myocardium (see Figure 2 in revised version).

-          Authors could provide a small/medium-size paragraph to current pharmacotherapeutic avenues that could limit scar formation/infarct size in the clinical setting as this would enhance the „clinical“ side of cardioprotection or non-dogmatic treatments that might influence these endpoints

Thank you for the suggestion. We added this information into the introduction.

-          Please consider references on, for example, ivabradine use in the ACS (Niccoli et al. PMID: 28256323; Priti et al. PMID 28423233; O'Connor et al. PMID: 27107133), granulocyte-colony stimulating factors (D'Amario et al. PMID: 28602846; Huang et al. PMID: 29651017), metoprolol (Garcia-Prieto et al. PMID 28416795; Pizarro et al. PMID: 24694530), etc.

References added with revision in point above.

-          Please check and fix references throughout the manuscript and at the References list as it seems some have missing information.

The references have been reviewed and corrected.

Reviewer 2 Report

The article in question is a good review on the subject. It includes a correct classification of the heart's ischemic pathology and an in-depth report of the main molecules potentially endowed with cardioprotective effect.

I would like to point out only some useful additions in my opinion:

1- line 60: "... are positively correlated with infarct size" and the segment involved, since the more or less probable dysfunction of the mitral valve also depends on the involved segment.

2- line 79: "... comparable to coronary artery surgery" in the short term, even if in the long term [1] and in particular pathological settings [2,3] coronary bypass shows a longer duration over time with lower need for repeated revascularization.

3- line 139: Limiting the extent of the initial infarct is also important because it is shown that the residual tissue maintains its three-dimensional structure useful for a possible surgical reconstruction [4].

Lastly, if the editorial rules of this newspaper allow it, I would use Figure 2 as a useful and informative graphical abstract of the article.

1- Mohr FW et al. Coronary artery bypass graft surgery versus percutaneous coronary intervention in patients with three-vessel disease and left main coronary disease: 5-year follow-up of the randomised, clinical SYNTAX trial. Lancet 2013 Feb 23; 381:629.

2- Y.G. Kim, D.W. Park, W.S. Lee, et al. Influence of diabetes mellitus on long-term (five-year) outcomes of drug-eluting stents and coronary artery bypass grafting for multivessel coronary revascularization. Am J Cardiol, 109 (2012), pp. 1548-1557

3- Ye Y, Yang M, Zhang S, Zeng Y. Percutaneous coronary intervention versus cardiac bypass surgery for left main coronary artery disease: A trial sequential analysis. Medicine (Baltimore). 2017;96(41):e8115

4- Cirillo M, Arpesella G. Rewind the heart: A novel technique to reset heart fiber orientation in surgery for ischemic cardiomyopathy. Med. Hypotheses 2008, 70, 848–854

 Author Response

Dear Reviewers:

              We appreciate the comments you provided and believe the changes made based on your comments will significantly strengthen the manuscript entitled, “Use of Multifactorial Treatments to Address the Challenge of Translating Experimental Myocardial Infarct Reduction Strategies.” We have addressed your comments point-by-point in Italics in the following letter and included a “Tracked Changes” revised version of the document.

Reviewer 2:

Comments and Suggestions for Authors

The article in question is a good review on the subject. It includes a correct classification of the heart's ischemic pathology and an in-depth report of the main molecules potentially endowed with cardioprotective effect.

I would like to point out only some useful additions in my opinion:

1- line 60: "... are positively correlated with infarct size" and the segment involved, since the more or less probable dysfunction of the mitral valve also depends on the involved segment.

We added language to address this.

2- line 79: "... comparable to coronary artery surgery" in the short term, even if in the long term [1] and in particular pathological settings [2,3] coronary bypass shows a longer duration over time with lower need for repeated revascularization.

We added language for clarification to address this point.

3- line 139: Limiting the extent of the initial infarct is also important because it is shown that the residual tissue maintains its three-dimensional structure useful for a possible surgical reconstruction [4].

We added language to address the additional benefit of limiting initial infarct size wisely noted by this reviewer. 

Lastly, if the editorial rules of this newspaper allow it, I would use Figure 2 as a useful and informative graphical abstract of the article.

 This is now Figure 3. We have no objection to using this Figure as a graphical abstract.

1- Mohr FW et al. Coronary artery bypass graft surgery versus percutaneous coronary intervention in patients with three-vessel disease and left main coronary disease: 5-year follow-up of the randomised, clinical SYNTAX trial. Lancet 2013 Feb 23; 381:629.

2- Y.G. Kim, D.W. Park, W.S. Lee, et al. Influence of diabetes mellitus on long-term (five-year) outcomes of drug-eluting stents and coronary artery bypass grafting for multivessel coronary revascularization. Am J Cardiol, 109 (2012), pp. 1548-1557

3- Ye Y, Yang M, Zhang S, Zeng Y. Percutaneous coronary intervention versus cardiac bypass surgery for left main coronary artery disease: A trial sequential analysis. Medicine (Baltimore). 2017;96(41):e8115

4- Cirillo M, Arpesella G. Rewind the heart: A novel technique to reset heart fiber orientation in surgery for ischemic cardiomyopathy. Med. Hypotheses 2008, 70, 848–854

Round  2

Reviewer 1 Report

I would wish to congratulate to authors for addressing all my concerns and suggestions effectively. This manuscript is now much improved and acceptable for publication.